# Getting personal with epigenetics: towards individual-specific epigenomic imputation with machine learning

Alex Hawkins-Hooker[1,2,3,5] ✉, Giovanni Visonà [2,5], Tanmayee Narendra [1,4], Mateo Rojas-Carulla[2], Bernhard Schölkopf[2] & Gabriele Schweikert [1,4] ✉

Epigenetic modifications are dynamic mechanisms involved in the regulation of gene expression. Unlike the DNA sequence, epigenetic patterns vary not only between individuals, but also between different cell types within an individual. Environmental factors, somatic mutations and ageing contribute to epigenetic changes that may constitute early hallmarks or causal factors of disease. Epigenetic modifications are reversible and thus promising therapeutic targets for precision medicine. However, mapping efforts to determine an individual's cell-type-specific epigenome are constrained by experimental costs and tissue accessibility. To address these challenges, we developed eDICE, an attention-based deep learning model that is trained to impute missing epigenomic tracks by conditioning on observed tracks. Using a recently published set of epigenomes from four individual donors, we show that transfer learning across individuals allows eDICE to successfully predict individual-specific epigenetic variation even in tissues that are unmapped in a given donor. These results highlight the potential of machine learning-based imputation methods to advance personalized epigenomics.

Epigenetic mechanisms play an essential role in developmental biology and human disease[1,2]. They act at the intersection of genetic and environmental factors to control, regulate, and propagate cellular responses, significantly contributing to diverse cellular phenotypes. Importantly, their influence on gene activity is reversible without altering the underlying DNA sequence. Therefore, they provide unique diagnostic and therapeutic opportunities and offer promising targets for precision medicine approaches[3–5], with particular interest in applications in cancer treatment[6,7]. Advances in epigenome editing technologies are paving the way for including epigenetic modifications not just as biomarkers, but also as direct intervention targets for novel treatments[8–10]. However, crucial challenges remain, mainly because epigenomes are cell-type specific and dynamically changing on different time scales, for example during the cell cycle, development, or ageing. Therefore, decoding epigenetic patterns is particularly laborious, expensive, and data-intensive.

A more in-depth understanding of epigenetic modifications has shed new light on the mechanisms involved in certain neurological and neurodegenerative diseases, developmental disorders, and some forms of cancer[3,11–13]. Large-scale efforts to map the functional properties of human epigenomes proved essential for these developments and have provided a crucial resource to understand how the interplay between genetic and epigenetic factors affects cellular identity and function[14,15]. While these projects aim to profile diverse cell types exhaustively using various epigenetic assays, the associated experimental costs impose constraints that lead to incomplete maps, with many cell types still sparsely analysed. This sparsity presents a particular challenge for the study of individual-specific epigenomic

[1]School of Life Sciences, University of Dundee, Dow Street, Dundee DD1 5EH, UK. [2]Empirical Inference Department, Max-Planck Institute for Intelligent Systems, Max-Planck-Ring 4, Tübingen 72076, Germany. [3]Centre for Artificial Intelligence, University College London, London, UK. [4]Interfaculty Institute for Biomedical Informatics, University of Tübingen, Sand 13, Tübingen 72076, Germany. [5]These authors contributed equally: Alex Hawkins-Hooker and Giovanni Visonà. ✉e-mail: alex.hawkins-hooker.20@ucl.ac.uk; G.Schweikert@dundee.ac.uk

variation. Individual-specific epigenomic signatures have the potential to inform personalized predictions for risk stratification[16], drug resistance[17,18], or personalized therapies[19], however producing comprehensive individual-specific epigenomic maps remains practically infeasible, not least because of the difficulty of obtaining samples from certain tissues.

As a result, computational approaches that can leverage existing epigenomic data to impute the results of as-yet unperformed assays are of considerable interest, particularly if they are able to predict individual-specific variation. As well as advancing overall understanding of the epigenomic landscape, effective imputation methods have the potential to play a role in the development of novel precision medicine workflows, for example by predicting the results of epigenetic assays in tissues that are difficult to probe in living patients, or aiding in the prioritization of epigenomic measurements[20]. Previous work in epigenomic imputation has shown that machine learning models can be trained to exploit the correlations between sets of epigenomic marks within and between cell types to successfully predict missing measurements[21-23]. However, these studies have focussed on the imputation of reference epigenomes, and have not explored the use of imputation methods to generate individualized predictions.

In this work, we introduce eDICE (Fig. 1a), a Transformer-inspired imputation model, which is trained to impute missing epigenomic tracks given sets of observed tracks. eDICE learns to encode the epigenomic signal in a set of observed tracks into factorised local representations of each cell type and each assay, enabling imputations to be made for unseen combinations of cell type and assay by decoding from the appropriate representations. We first show that our architecture leads to improved imputation performance relative to previous methods on the reference Roadmap epigenomes while conveying significant practical benefits. Next, we use recently published individual-specific epigenomes from EN-TEx[24] to test whether eDICE can be used to generate individualized epigenomic imputations. Inspired by precision medicine applications, we devise a task designed to assess the utility of imputation methods for predicting individualized epigenomes in hard-to-access tissues and find that transfer learning across individuals allows eDICE to predict individual-specific epigenomic variation in this setting.

## Results

### eDICE and previous work on epigenomic imputation

In 2015, Ernst and Kellis pioneered work in the field of large-scale epigenomic imputation by introducing ChromImpute[21], an imputation strategy for the reference epigenomic datasets produced by the Roadmap and ENCODE projects[15,25]. Given sets of reference epigenomes generated by performing various epigenomic assays in a set of cell-types, the epigenomic imputation task posed by ChromImpute is that of predicting epigenomic tracks representing combinations of cell-type and assay for which experimental data are not available, thereby 'completing' the epigenomic map. To solve this problem, ChromImpute adopts a regression-based approach, requiring the training of a separate ensemble of models for each target track. While ChromImpute has shown effective performance, it relies on the manual engineering of input feature sets and the training of thousands of separate models, preventing the effective sharing of information across the highly related tasks of the imputation of different tracks.

Subsequently, imputation strategies based on tensor factorization have been proposed as a way of reducing the complexity of ChromImpute. PREDICTD[22] generates predictions via a linear combination of learned factors representing cell type, assay, and genomic location. Avocado[23] replaces the linear combination of factors used in PREDICTD with a learned nonlinear operation, by passing concatenated embeddings corresponding to each factor through a neural network. Tensor factorization approaches have the appealing property that given a learned set of factors, predictions can be generated at any

genomic location, for a track corresponding to any combination of one of the modelled cell-types and assays. Nonetheless, the performance of these approaches has only outstripped ChromImpute on a subset of metrics.

Seeking to combine the strengths of prior approaches, we developed a deep learning model, eDICE (*epigenomic Data Imputation via Contextualized Embeddings*), based on framing the epigenomic imputation problem as one of masked input reconstruction. During training, a random subset of the observed signal values for a set of epigenomic tracks at a single genomic position is masked out, and the model is tasked with learning to impute the masked values given the remaining observed values. Unlike in standard masked input reconstruction applications, the epigenomic imputation problem requires models to be capable of predicting signal values for tracks never seen during training, representing novel combinations of cell type and assay. To achieve this combinatorial form of generalization, eDICE encodes the input signal at the genomic position of interest into separate latent representations summarizing the local epigenomic state of each cell type and the local activity profile of each assay. The signal value in a masked track is then reconstructed by concatenating the representations for the relevant cell type and assay and passing them through a Multi-Layer Perceptron (MLP) decoder. At test time, predictions for new tracks can be generated in the same way, by feeding the model with the signal values of a set of observed tracks at a genomic location of interest, and decoding from the representations of the target cell type and assay.

To implement the factorized encoding of local epigenomic signal, we developed a self-attentive neural network module (Fig. 1), based on the Set Transformer architecture[26]. This module starts by independently encoding the signal in each cell type and each assay, then constructs 'contextualized' representations of each cell type and each assay by transferring information among related cell types and related assays using self-attention. By conditioning on observed signal values to build representations of the local epigenomic state, rather than learning location-specific embeddings, eDICE achieves the generalization capacity of tensor factorization models while offering substantial improvements in both training efficiency and performance. A full description of the architecture and training procedure is provided in the Methods section.

### eDICE imputations are highly accurate on the reference epigenomes

For direct comparison with previous imputation work, we evaluated the accuracy of eDICE imputations on a dataset of epigenomic tracks collated by the Roadmap project[27] and used in previous studies[21-23]. This dataset consists of 1014 signal tracks from 24 epigenomic assays in 127 cell types. All but one of the assays target histone modifications, with the remaining assay profiling chromatin accessibility via DNase-seq. A core set of five assays (H3K4me1, H3K4me3, H3K36me3, H3K27me3 and H3K9me3) is available in most cell types, while coverage of the cell types with the remaining assays varies widely. We used the first train/test split defined by Durham et al.[22], which consists of 709 training tracks, 102 validation tracks, and 203 test tracks (Supplementary Fig. 1 and Supplementary Section 3.1). To compare the performance of imputation methods, we report a series of metrics assessing the quality of imputations of the tracks in the test set by models trained on tracks in the training and validation sets (and optionally using these tracks to provide inputs at test time). The metrics are computed across chromosome 21 of the hg19 assembly, the smallest human chromosome, spanning approximately 48 million base pairs. As baselines, we report results for the prior methods ChromImpute, PREDICTD, and Avocado (see the Methods section for further details). Finally, as a parameter-free baseline, we also report predictions made by averaging the signal of the target assay in all other cell types in the training data except the target cell type (AVG).

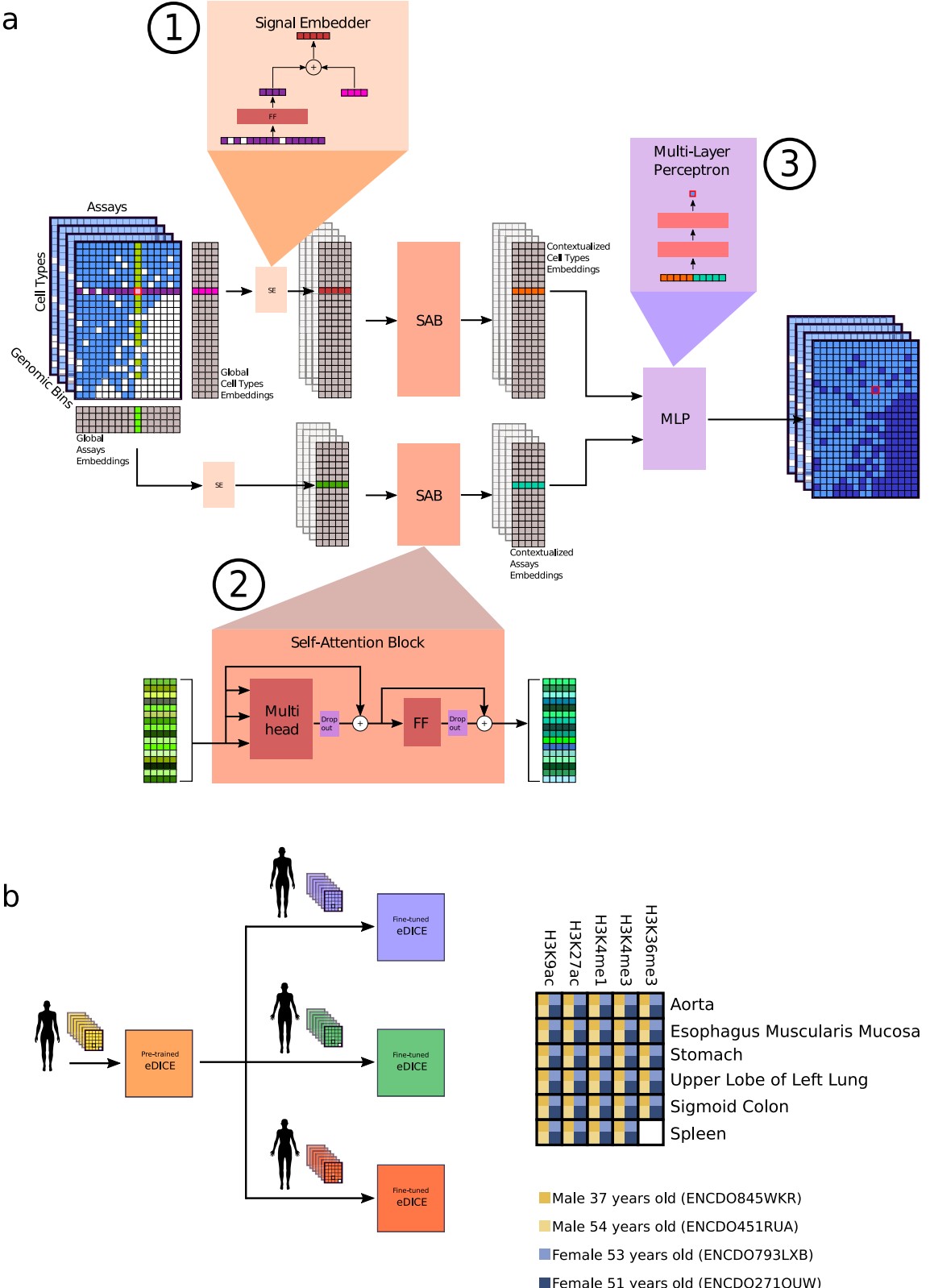

**Fig. 1 | Schematic representation of the eDICE model. a** For each cell type, we collect all measured signal values from assays performed in that cell type at the target bin, and project this set of values into a shared embedding space, where it is combined with a global embedding representing the cell type (1). We do likewise for assays, projecting the sets of values measured in different cell types from each assay into a distinct embedding space. We then apply self-attention over both sets of embeddings, allowing the network to capture relationships between cell types and assays to produce `contextualized' latent embeddings which are functions of the local signal values in all observed tracks (2). Finally, a feed-forward neural network combines the contextual embeddings for a target cell type-assay combination to generate a prediction for the local signal value (3). **b** Transfer learning scheme for the imputation of unseen tissues in the EN-TEx dataset.

Previous studies of imputation methods have varied in the choice of the primary metrics by which to assess performance[21–23]. In an attempt to provide a balanced view of model quality, we report performance on a selection of metrics designed to capture three desirable characteristics of imputations: (i) global similarity between imputations and ground truth values (ii) similarity between imputations and ground truth values focusing on foreground (Fg) and background (Bg) bins, as determined by MACS2[28] and (iii) discriminative accuracy for a peak vs non-peak classification task. The first two categories are assessed using mean-squared error and Pearson correlation on the arcsinh-transformed signal values, while in the latter category, we measure peak classification performance via the threshold-agnostic area under the precision-recall curve (AUPRC), as well as by precision and recall after calling peaks on the imputations using MACS2. Additional details on all metrics are found in Methods and Supplementary Section 3.2.

The performance of the models is presented in Fig. 2 and Supplementary Figs. 2–9, with numeric values reported in Supplementary Table 2. eDICE outperforms PREDICTD and Avocado across all metrics, and ChromImpute across the majority, although ChromImpute shows strong performance for the prediction of peak height in the foreground (Fig. 2a, c, Supplementary Figs. 2, 3). eDICE's relative disadvantage here suggests a tendency to systematically underestimate the absolute signal values within peaks, which is exemplified in the trade-off between precision and recall compared to ChromImpute. However, it ranks peak and non-peak regions relative to each other more accurately than ChromImpute, as demonstrated by the fact that it outperforms all baselines on the AUPRC metric, thereby offering the best overall imputation in terms of global discriminatory power. We emphasize that while PREDICTD and Avocado generated imputations respecting the same data split used for eDICE, the ChromImpute imputations were produced in a leave-one-out fashion, so our model's improved performance comes despite a considerable handicap relative to ChromImpute in terms of the available training data.

Qualitatively, eDICE presents many of the characteristics that were present in its predecessors, such as a general smoothing of the imputed tracks, which is especially notable in the background regions (example in Fig. 2b). Additionally, the imputed tracks reduce the impact of outlier values, such as the extremely high peaks present in a few tracks for H3K4me3. Such peaks are not necessarily a direct representation of the high significance of the local enrichment but can be heavily affected by the control samples' coverage and quality, which, when low, can bias the estimated p-values towards extreme values (see further discussion in Supplementary Section 3.1).

To confirm that these aggregate results were not unduly influenced by variation in the range of metric values across different types of assay, we also examined the metrics at the level of individual tracks (Fig. 2c and Supplementary Figs. 4–6) and aggregated by assay (Fig. 3a). The track-level comparisons confirm that eDICE's performance improvements are consistent across different combinations of cell-type and assay. Grouping tracks by assay reveals significant differences in the performance of imputation methods depending on the type of epigenetic mark. For example, all models tend to perform relatively poorly when predicting H3K27me3 and H3K9me3 (Fig. 3a and Supplementary Figs. 7–9). Comparing the average assay-level performance of each model shows that the improvements brought by eDICE are consistent across the board despite these discrepancies between assays (Supplementary Fig. 2).

Finally, we explored whether differences in performance between types of assays could be related to differences in specific properties of the epigenetic marks. Some histone modifications can be classified as either narrow-peak (H3K27ac, H3K4me2, H3K4me3, H3K9ac) or broad-peak marks (H3K27me3, H3K36me3, H3K4me1, H3K79me2, H4K20me1). Comparing the performance of eDICE on test tracks across these two groups, we observed that performance tended to be higher on narrow-peak than on broad-peak marks for correlation and classification metrics (Fig. 3b). Furthermore, a similar divide is observed when splitting histone modifications into repressive (H3K27me3, H3K9me3) and activating marks (the active promoter-associated H3K9ac, H3K4me2, H3K4me3, active enhancer-associated H3K4me1 and H3K27ac and DNAse-seq, Fig. 3c). As repressive marks are often linked to heterochromatin configurations, this discrepancy is possibly due to biases introduced by the processing pipelines because of systematic sequencing differences in these regions. However, as repressive marks also tend to display broad peaks, it is challenging to pinpoint the precise reason for the observed differences.

Importantly, the performance benefits of eDICE are coupled with increased efficiency in the training procedure. Relative to ChromImpute, this is a result of training only a single model rather than a separate ensemble of models for each target track. Relative to Avocado and PREDICTD, eDICE can make accurate genome-wide predictions without needing to train on every genomic location, leading to major improvements in training efficiency. To highlight this, in Fig. 2d, we show the results of training eDICE on smaller subsets of the randomly selected genomic locations. Even when trained on a small fraction of the available genomics data, eDICE outperforms Avocado, suggesting that the tensor factorization models severely overparameterize the imputation problem by learning a representation for each genomic bin.

## Imputations capture significant differences between tissues

Epigenomic patterns differ between cell types to control and register cell function and identity. It is critical that imputations accurately capture these differences if they are to constitute valuable resources of cell-type-specific epigenomic landscapes. However, within the scale of the whole genome, these cell-type-specific differences are subtle and global evaluation metrics such as those considered above are dominated by regions that have a shared functionality across cell types, such as large intergenic regions.

In the analysis of epigenetic modifications, it is crucial to capture not just a single instance of the local signal measured by experimental assays, but also the local variability which may characterize each tissue. To distinguish potentially functional differences from either technical or biological fluctuations, established experimental protocols explicitly require several biological replicates to estimate local variability. This is essential for robust statistical hypothesis testing[29]. On the other hand, experimental tracks are generally pooled for imputation tasks, and predictions thus constitute mean epigenomic tracks per cell type, where the inherent variability is lost. We present here a case study in which we estimated local variability on the training data and then generated simulated replicates from the mean epigenomic imputation. This strategy was used to predict and identify differential peaks in H3K9ac tracks across two tissues (corresponding to Roadmap cell types Adipose-Derived Mesenchymal Stem Cell Cultured Cells (E025) and Muscle Satellite Cultured Cells (E052)).

We highlight that the overall shape of individual peaks is remarkably conserved between individual experimental replicates for corresponding tracks (Fig. 4a). In the case of tissue-specific peaks, on the other hand, the signal shapes are distinct between replicated measurements derived from different tissues (Fig. 4b). We have previously exploited this observation for differential peak calling[30], where we considered the genomic region of the peak as a metric space and treated the pile-up of sequenced reads like a sample from a hidden probability distribution on that space. This strategy dramatically improves the test's statistical power compared to methods based on total counts alone. We also note that the shape differences are well captured by the mean signals (Fig. 4a, b bottom panels). To quantify differences in peak shapes across the two cell types, we computed the Wasserstein (WS) distance between the pooled ground truth signals across the two cell types, and

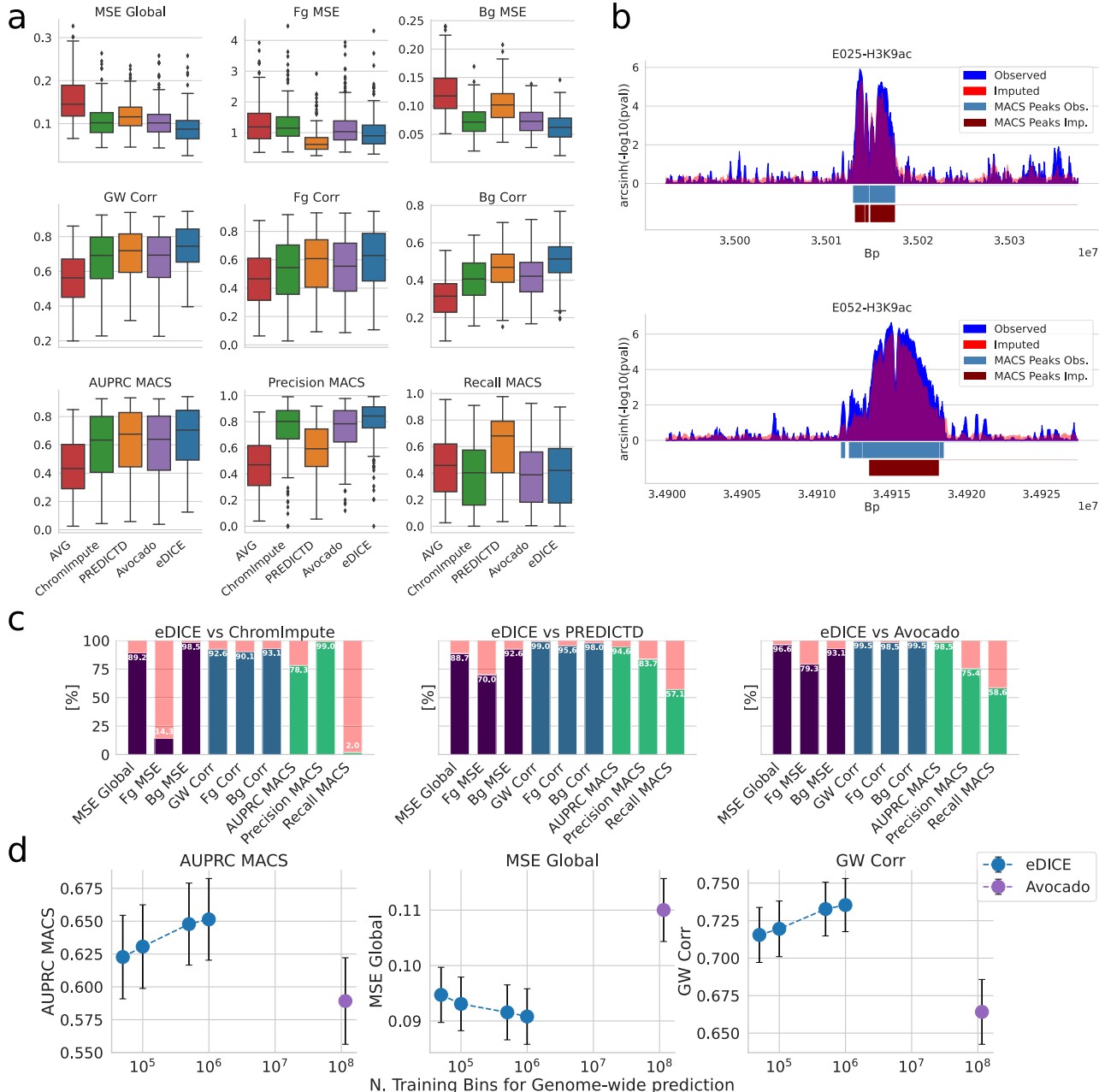

**Fig. 2 | Comparison of imputation methods on the Roadmap reference epigenomes. a** Performance metrics for the imputation of the n=203 test tracks on chromosome 21 for each model. Boxes represent the interquartile range (IQR), with the middle line representing the median; the whiskers represent points that lie within 1.5 IQRs of the lower and upper quartiles while remaining outliers are explicitly displayed. Metrics presented include Mean Squared Error (MSE) and Pearson correlation coefficient (Corr) for the Genome-wide (GW/Global), Foreground (Fg) and Background (Bg) regions, as well as the Area Under the Precision-Recall Curve (AUPRC), Precision, and Recall for the classification of peaks detected with MACS2. **b** Examples of observed epigenomic tracks with the signals imputed by eDICE for the assay H3K9ac in two selected tissues (E025, E052). Below the tracks, the peaks detected with MACS2 highlight how the imputations accurately capture enriched regions. The peaks were detected using a one-sided Poisson hypothesis test with Benjamini-Hochberg correction for multiple test corrections and a cut-off value of 0.01. **c** Percentages of test tracks on which eDICE outperforms the baselines for each metric. ChromImpute shows good performance on tasks related to the height of the peaks, while eDICE outperforms PREDICTD and Avocado on all metrics. **d** Learning curves that display several global performance metrics against the number of genomic positions used in training. Tensor factorization models such as Avocado need to be trained on the whole genome to make genome-wide predictions. eDICE, on the other hand, can be trained efficiently on a small subset of genomic regions and still obtain improved performance, suggesting that previous models severely overparameterized the imputation problem. Data are presented as mean ± 95% confidence interval for $n = 203$ test tracks. Source data are provided as comma-separated-values (csv) files.

likewise between the imputed signals. Figure 4c shows that the distances in imputed and ground truth tracks are strongly correlated, indicating that the imputations accurately capture cell-type-specific differences in the shape of signal enrichment at peak regions.

As an independent analysis, we next took advantage of the robustness of the existing differential peak analysis method, DiffBind[31]. Since DiffBind requires replicates for statistical testing, we estimated the local variability of cell-type-specific test tracks (Supplementary Section 3.4). Assuming a negative binomial distribution, the estimated

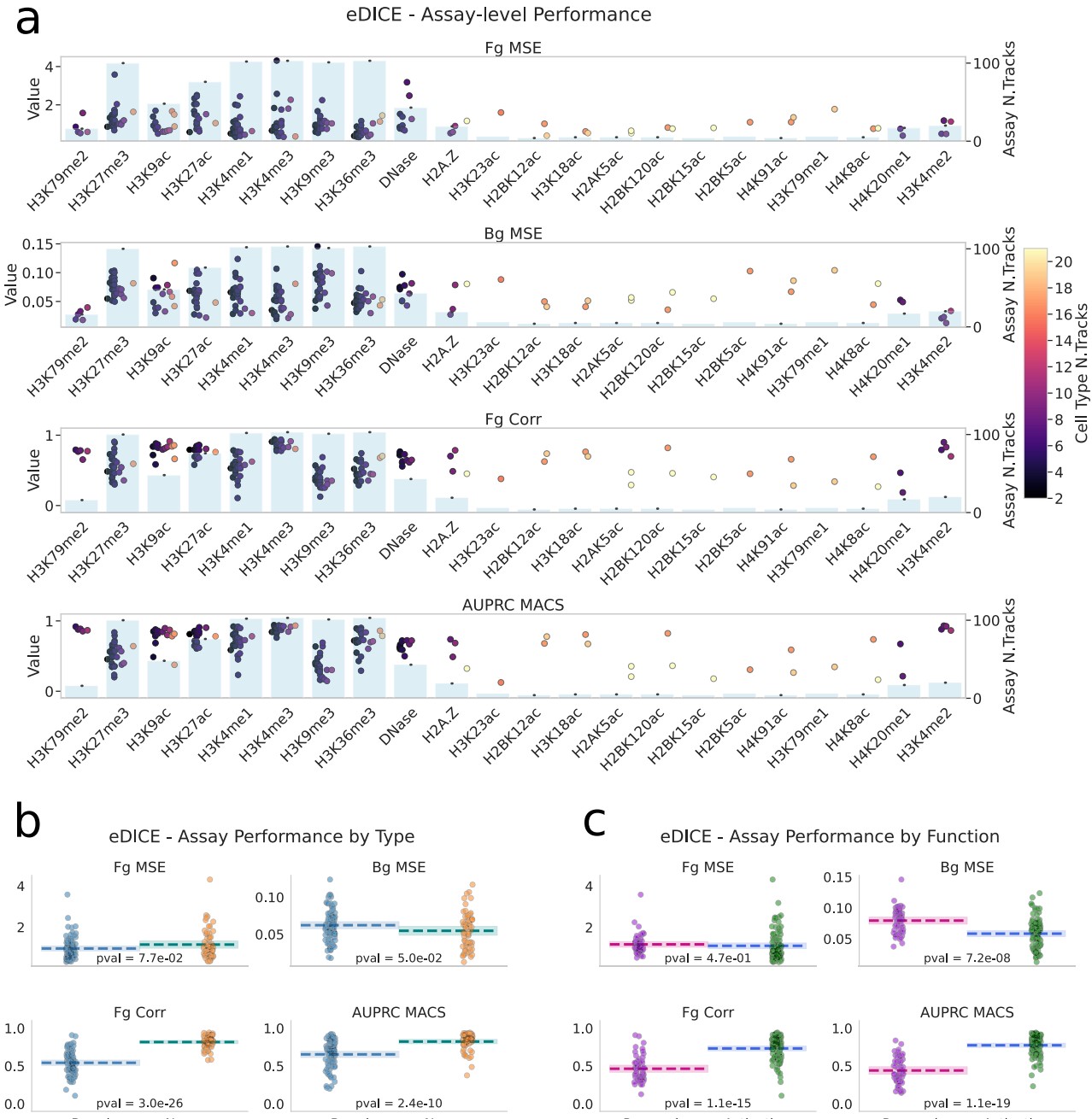

**Fig. 3 | Imputation performance varies significantly between different assays. a** Grouping the tracks by assay reveals considerable differences in the imputation performance. This phenomenon is observed in the previous models as well, indicating that it is most likely due to the nature of the specific modifications and the biases that their signal includes. The colour of each dot indicates the number of training tracks that share the cell type with that specific test track, while the light blue bars in the background show the number of training tracks that share the same assay. **b** Assays split into broad- and narrow-peak marks show consistently different

performance for the imputation task. For each metric, we performed a 2-sided Welch's t-test under the null hypothesis that both sets of metrics have the same mean and reported the resulting p-value at the bottom of each plot. **c** Splitting the histone marks by functionality (repressive vs. activating) shows a similar bias as the comparison in (**b**). For each metric, we performed a 2-sided Welch's t-test under the null hypothesis that both sets of metrics have the same mean and reported the resulting p-value at the bottom of each plot. Source data are provided as comma-separated-values (csv) files.

variance parameters were subsequently used to simulate replicates from the imputed mean signal tracks on chromosome 21. While an arbitrary number of replicates can readily be generated in this way, we chose to use three to four simulated replicates, similar to typical experimental scenarios. Those tracks were fed into the standard differential analysis pipeline, and the outcome was compared with the results obtained from the corresponding analysis of actual replicated measurements. We emphasize that the simulation procedure

employed only replicates from the training set and tissue-specific control samples in addition to the imputed tracks, and made no use of any information from the test set.

Employing the DiffBind library[31] we compared *binding affinity scores*, which are indicative of the strength of interaction between DNA and biomolecules (such as modified histones). Figure 4d shows a correlation heatmap for the similarity of affinity scores for different samples. The block structure highlights the expected relationship

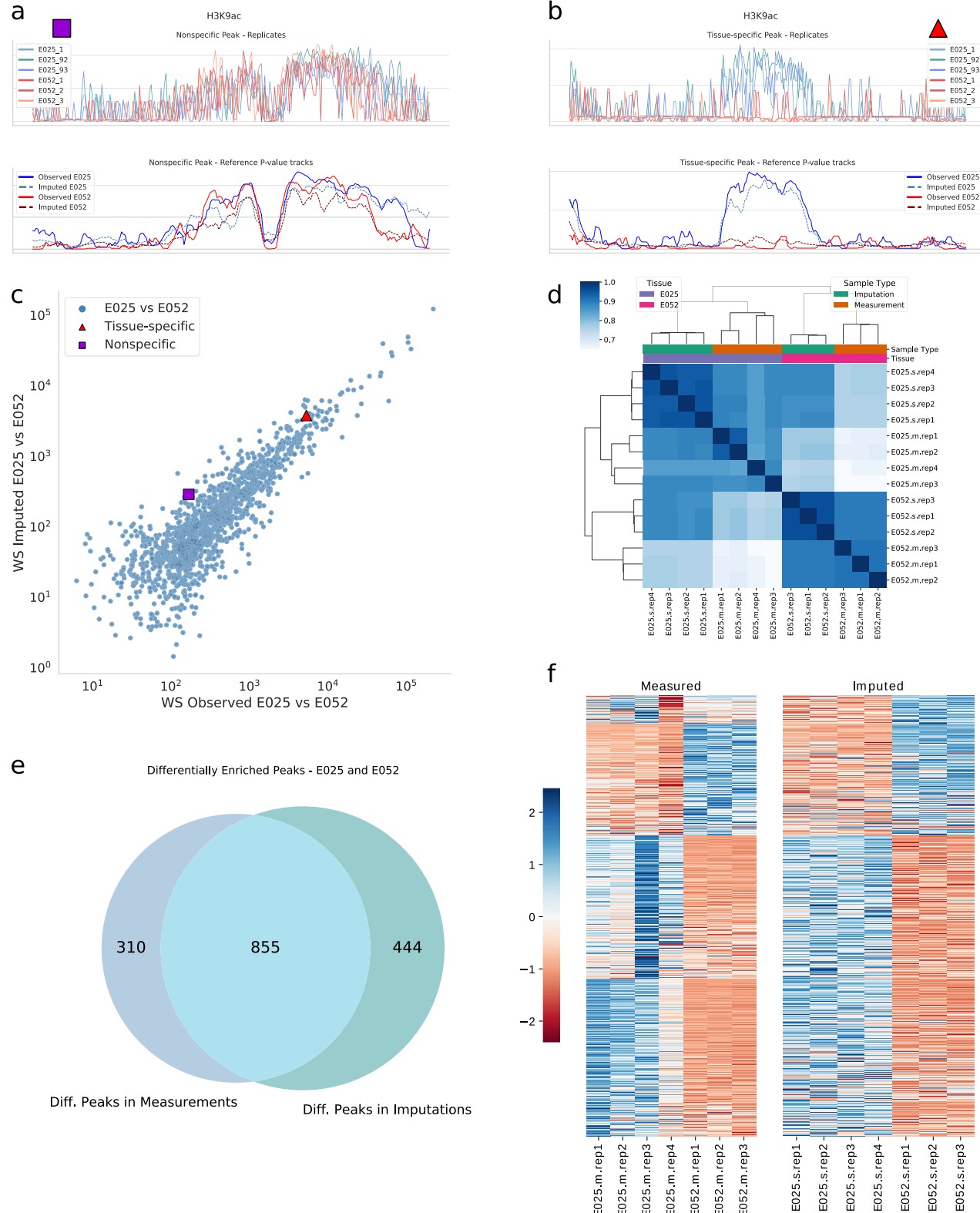

between the replicates derived from different tissues; however, the simulated replicates show high similarity across tissues, possibly due to the adopted procedure underestimating the biological variance between samples.

Within DiffBind, we used DESeq2 to identify peaks of differential enrichment with default parameters. Specifically, we used a 'glmGamPoi' fit type to estimate dispersion and used a Wald test for

negative binomial distribution ('nbinomWaldTest') to identify statistically significant peaks. A total of 1165 and 1299 peaks were detected as differentially enriched in measurement and imputations, respectively (FDR threshold of 0.05). 855 peaks (~73% of the measured peaks) are shared between the two sets, resulting in a Positive Predictive Value of 0.66 (Fig. 4e). Binding affinity scores for each differentially enriched peak in the consensus peak set derived from imputations and

**Fig. 4 | Differential peak analysis using imputed epigenomic tracks. a, b** show examples of non-specific and tissue-specific peaks respectively for H3K9ac in the two chosen tissues (E025 and E052). The upper part shows the measured replicates, while the lower portions display the aggregate p-value tracks for the observations and the corresponding imputations. The aggregate tracks do not capture the information on the biological variability between samples. **c** A scatter plot of the Wasserstein distance between the signal in the two tissues, for each peak in the enriched peakset of E025. The x-axis displays the WS distance between observed signals, while the y-axis is between imputed signals. The imputations retrieve most of the information contained in the measurements, especially for the stronger differences between tissues. We highlighted the two points corresponding to the peaks shown in (**a**) and (**b**). **d** Correlation heatmap of the affinity scores for different replicates. The simulated replicates correctly retrieve the expected relationships to the measured replicates, although they show a high degree of similarity between themselves, likely an artefact of the simulation procedure. **e** Venn diagram representing the peaks that are detected as differentially enriched between tissues using imputed and measured signals. The imputed signal retrieves 66% of the true peaks. **f** Binding affinity heatmaps for the measured replicates and the imputed pseudo-replicates. Each row corresponds to one of 1609 differentially enriched peaks detected in either of the measurements and imputations. The imputed replicates display the same global block structure as the measurement replicates. Source data are provided as comma-separated-values (csv) files.

measurements are shown in Fig. 4f, where the block structure resulting from agglomerative clustering of the measurements (left side) is replicated in the imputations (right side).

The differential analysis procedure was repeated for all the models analysed, with eDICE outperforming Avocado and ChromImpute; the model PREDICTD showed comparable performance (Supplementary Fig. 23). In summary, we conclude that the imputations accurately capture cell-type-specific differences, both in terms of altered shapes of signal enrichment at peak regions and also regarding integrated total counts in the peak regions, when considering local variability. In general, increasing the number of replicates in a sequencing experiment leads to more robust results[32]. Therefore, a similar augmentation strategy could also be applied to complement certain existing experimental data sets with additional replicates from an imputed mean track.

## eDICE accurately predicts personalized epigenomes in unseen tissues

Recent advances have highlighted the role that alterations in the epigenetic machinery play in human disease[33–35]. In the field of precision medicine, epigenetic mutations are currently examined mainly for their potential role in early detection and drug response prediction[36–38]. However, increasingly robust epigenome editing methods[8] open up exciting opportunities for direct interventions on the epigenome for the treatment of illnesses such as cancer[10]. Achieving a more in-depth understanding of individual- and cell-type-specific epigenetic patterns and their effect on the cellular machinery will be crucial to realizing the promise of such applications.

Recently, a collaboration between the ENCODE[25,39] and the Genotype-Tissue Expression (GTEx) consortia created data sets that include extensive individual-specific histone modification measurements from four donors[24]. We decided to use this dataset to test whether eDICE could be applied to impute epigenomic measurements in an individual-specific manner. One particular use case for imputations in this setting could be to predict epigenomic measurements in otherwise hard-to-access tissues, potentially avoiding the need for invasive procedures. Motivated by this use case, we developed a task to test the prediction of epigenomic measurements in a particular individual in tissues for which no epigenomic information for that individual is available. Specifically, we aim to impute epigenomic tracks for a target tissue in one individual patient ("target individual"), by using other observations from the same individual, as well as a more complete set of observations for another individual ("training individual"), which include the target tissue. To adapt eDICE to this task, we adopt a transfer learning approach. We first train an eDICE model on the complete set of observations for the training individual. The model is then fine-tuned on the set of observations for the target individual that do not include the target tissue, before imputing the target observations (Fig. 1b). We employed an eDICE model with the same architecture as used for Roadmap, but altered the masking process used during training to reflect the tissue-based prediction task, ensuring that the set of masked tracks at each genomic bin all belonged to a single randomly selected tissue.

To get a better understanding of this task, we performed an initial analysis of epigenomic variation between individuals in the EN-TEx dataset. The data include measurements spanning 25 different tissues from two adult males, 37 and 54 years old, and two adult females, 53 and 51 years old. We selected for further study 29 tissue-assay combinations comprising measurements of histone modifications available for all four individuals (Supplementary Table 3), focussing on chromosome 21 in all cases. Initial analysis of observed tracks revealed both a large degree of similarity in epigenomic signal across individuals in numerous instances (Fig. 5a) as well as the dominant role of tissue identity in determining epigenetic patterns, in particular for marks H3K27ac, H3K4me1 and H3K9me3 (Fig. 5b). Individual-specific peaks unique to only one or a subset of individuals are nonetheless observed, most notably for H3K9me3 (Fig. 5c). Three-dimensional histograms of co-occurrences across tissues and individuals highlight that across all marks individual-specific peaks are typically also specific to one or a small number of tissues and that the frequency of such peaks varies substantially between marks (Fig. 5d and Supplementary Figs. 17–21). These personal epigenomic differences may either reflect underlying DNA sequence variants, in which case they may be observable across different tissues of the same individual, or they may result as a consequence of ageing or due to interactions with external stimuli, potentially in a tissue-specific manner.

We next assessed the accuracy of eDICE imputations generated using the transfer learning scheme described above. We compared these predictions to imputations from two model-free baseline methods. The first method directly uses the corresponding track in the training individual as a prediction of a given track in a target individual. The second method generates a predicted track by averaging the tracks from the target assay in all tissues apart from the target tissue in the target individual (i.e., an individualized version of the AVG baseline used previously). For each method, we consider all possible combinations of target tissue, target individual, and training individual, and evaluate the resulting predictions. Using the transfer learning strategy presented, eDICE produces imputations which are globally more accurate than either of the baseline methods, as measured by MSE and Pearson correlation (Fig. 5e), indicating that transfer learning successfully adapts eDICE to the context of a new individual, while retaining the understanding of tissue types inherited from the training individual to allow successful prediction in tissues without measurements in the target individual.

## eDICE captures epigenetic variation between individuals

Finally, we used the same transfer learning framework to assess eDICE's ability to predict individual-specific epigenomic signatures. Defining such signatures is far from trivial; a robust analysis would require more than four individuals to properly understand the overlap of enriched regions and the external factors that influence them. As a working approximation, we define individual-specific peaks as those enriched regions detected from the measured samples that span at least 150 bp (i.e., the approximate length of the DNA wrapped around a nucleosome) and which are present only in the target individual, and not the training individual. This definition aims to capture peaks such

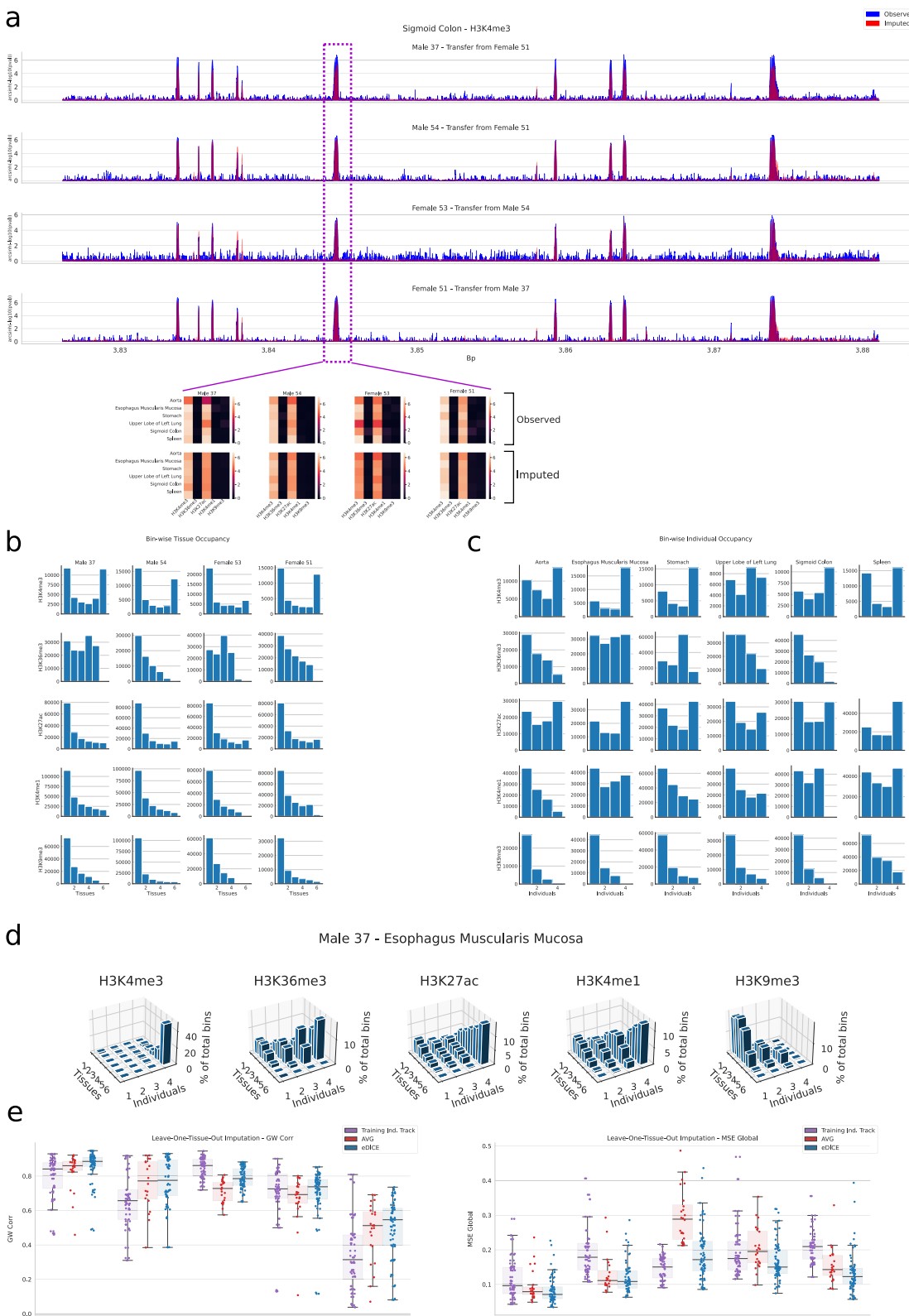

**Fig. 5 | Imputation of individual-specific epigenomic patterns. a** Sigmoid Colon-H3K4me3 track spanning 800kb and showing consistent patterns for all four individuals. For the central peak, we display a slice across the epigenomic tensor demonstrating signal conservation across tissues and individuals. **b** Occupancy histograms for the enriched bins across tissues. **c** Occupancy histograms for the overlap of enriched bins across individuals. **d** Occupancy across tissues and individuals for each enriched bin in the tracks for Male 37 for the Esophagus Muscularis Mucosa tissue. **e** Pearson correlation and MSE for the leave-one-tissue-out

imputation of chromosome 21 using transfer learning from one training individual to the target individual. $n = 72$ imputed tracks for the `Training Ind. Track' baseline and eDICE for each assay except H3K36me3, where $n = 60$. $n = 24$ for the "AVG" predictor for each assay except H3K36me3, where $n = 20$. Boxes represent the IQR, with the middle line representing the median; the whiskers represent points that lie within 1.5 IQRs of the lower and upper quartiles. Source data are provided as comma-separated-values (csv) files.

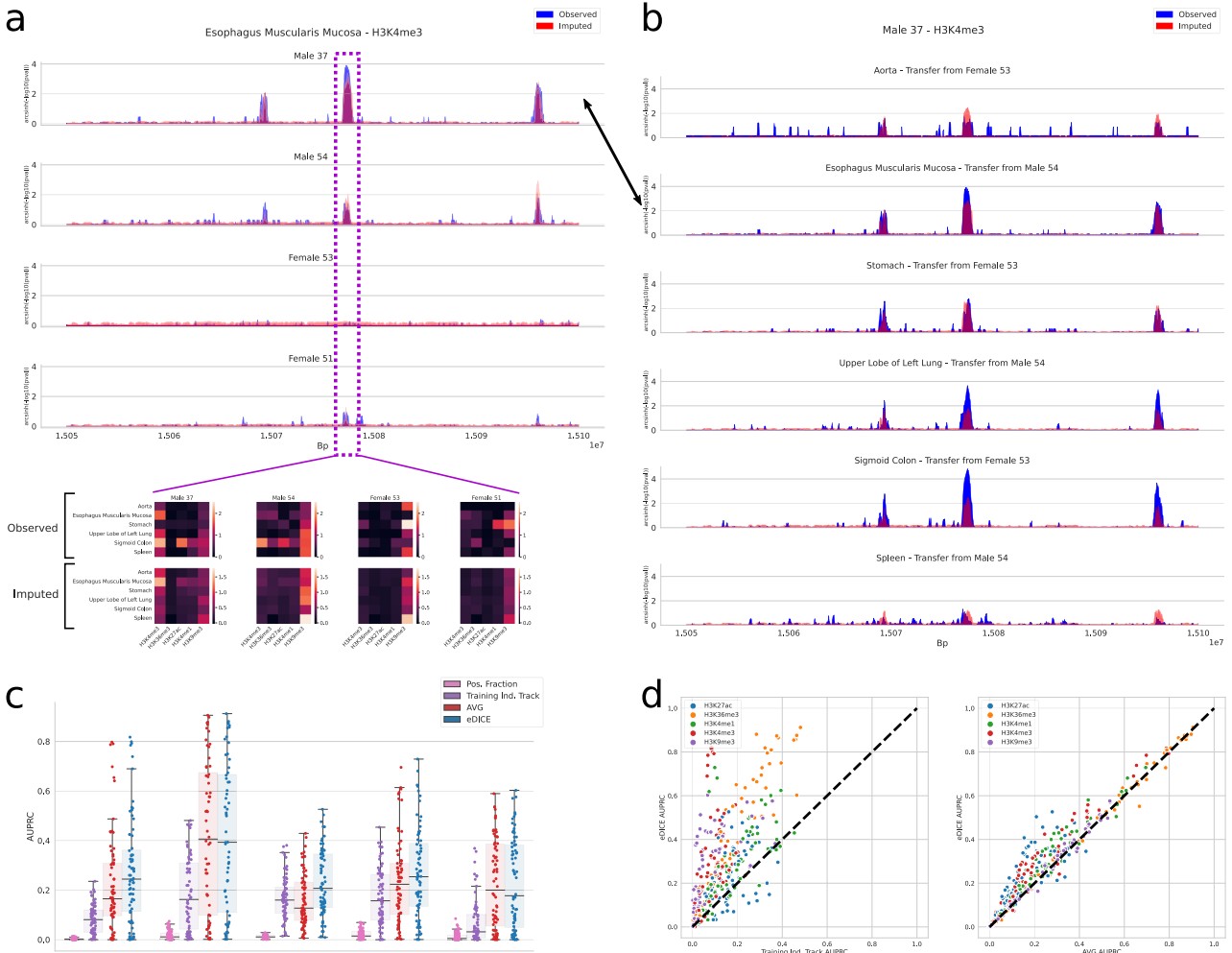

**Fig. 6 | Imputation for precision epigenomics. a** Individual-specific H3K4me3 enrichment for (Male 37) in Esophagus Muscularis Mucosa tissue. For this genomic location, we display a slice of the epigenomic tensor for each of the four individuals, highlighting the challenge of imputing these varied patterns. Peaks were detected with MACS2 using a one-sided Poisson hypothesis test with Benjamini-Hochberg correction for multiple test corrections and a cut-off value of 0.01. **b** Observed and imputed tracks for the H3K4me3 assay in Male 37 across tissues in the same genomic region as (**a**). **c** AUPRC for the prediction of individual-specific

enrichment in the LOO EN-TEx imputations, where the peaks shared with other individuals have been masked out. $n = 72$ imputed tracks for each model for all assays except H3K36me3, where $n = 60$ for each model. Boxes represent the IQR, with the middle line representing the median; the whiskers represent points that lie within 1.5 IQRs of the lower and upper quartiles. **d** Track-level AUPRC for the prediction of individual-specific enriched bins. Source data are provided as comma-separated-values (csv) files.

as the example shown in Fig. 6a, where H3K4me3 is clearly found in one individual. This task presents significant challenges due to the relatively small portion of epigenetic enrichments that meaningfully differ between individuals and because of the complex epigenetic patterns that arise in these regions of variability, exemplified by the heatmaps displayed in the lower portion of Fig. 6a. In these cases, local variability is observed not just between individuals, but also between tissues within the same individual (Fig. 6b).

To assess the ability of eDICE to predict these individual-specific peaks, we compared imputations to test tracks after excluding enriched regions shared with the training individual. We additionally excluded enriched regions specific to the test individual but spanning less than 150 bp. Within the remaining regions, we assessed the extent to which the imputations successfully distinguished individual-specific peaks from the background using the area under the precision-recall curve (AUPRC). For completeness, we included the fraction of positive samples (Pos. Fraction) as a standard baseline for the AUPRC measure[40]. The results, presented in Fig. 6c, show that eDICE improves the prediction of individual-specific enrichment compared to the

model-free baselines. A track-level comparison of eDICE's improvement over the model-free baselines is shown in Fig. 6d.

Capturing individual-specific differences is crucial for the robust application of state-of-the-art machine learning models to epigenetics within a clinical context. The case study presented aims to be a guiding example for the development of better and more accurate models that may be included in a clinical workflow.

## Discussion

We presented eDICE, a deep-learning-based epigenomic imputation framework which achieves high accuracy by combining the advantages of its predecessor models. Like ChromImpute, eDICE uses the local signal of observed tracks to encode information on the genomic position, removing the need to learn explicit embeddings for each position. Similar to the tensor factorization models PREDICTD and Avocado, eDICE uses factorized representations to achieve combinatorial generalization, while drastically reducing the required parameter count (Supplementary Table 1). On reference epigenomes, eDICE's performance is robust across a variety of metrics capturing different

facets of imputation performance, surpassing all baselines across the majority of metrics, while offering significant practical benefits as a simple single-model approach that is efficient to train and run.

We emphasize the need for imputation models to be trained and designed with the aim of including imputations in established bioinformatics processes. As a case study, we explored the possibility of simulating biological replicates from the imputed data, which are then used for differential peak calling obtaining results compatible with the measured replicates. We pose that future developments in the field of epigenomic imputation should account for and predict not only the average value of measurements but also the intrinsic biological variability of different samples. Explicitly modelling the variance of epigenomic measurements would allow for a more robust analysis to distinguish the differences caused by fluctuations due to the natural variability of the samples from the true differences between tissues and marks that encode the functional variations of cell profiles.

Finally, we demonstrated the possibility of imputing personalized epigenomic tracks, showing how eDICE can be adapted to generate imputations for unseen tissues that outperform those from model-free baselines. The transfer learning approach adopted allows the model to learn representations for all tissues from a training individual, which can then be transferred to a target patient, enabling accurate imputations in tissues where no data for the target individual is available. While our results offer a proof of concept for the direct applicability of methods originally designed for the imputation of reference epigenomes in this setting, individual-specific imputation presents additional challenges which future works might seek to address directly. In particular, further enhancements in accuracy might be unlocked by incorporating information from DNA sequences, as well as information aggregated across other individuals and from reference datasets to augment the relatively limited number of measurements in any single individual. In order to fully leverage the promise of transfer learning across epigenomic datasets, further consideration should be paid to the role of systematic biases introduced by differences in experimental methodology and bioinformatics pipelines used to process sequencing data, as highlighted by the findings of the ENCODE imputation challenge[41], to which we submitted a prize-winning entry using a predecessor of eDICE. We believe that our results offer a strong indication that machine learning methods are well-placed to address these challenges, and, in so doing, to help overcome the experimental constraints that limit our understanding of epigenetic variation.

## Methods

### Data

A dataset of epigenomic measurements from the Roadmap Consortium[15] was selected to allow direct comparison with prior imputation methods. The Roadmap dataset consists of 1014 signal tracks from 24 types of epigenomic assay in 127 cell types. Each signal track is obtained by mapping a set of sequence reads to a genome to form a genome-wide activity profile. All but one of the assays target histone modifications, with the remaining assay profiling chromatin accessibility via DNase-seq. A core set of five assays, targeting H3K4me1, H3K4me3, H3K36me3, H3K27me3 and H3K9me3, is available in each cell type, while coverage of the cell types with the remaining assays varies widely. We use the first train/test split defined by[22], which consists of 709 training tracks, 102 validation tracks, and 203 test tracks. Supplementary Fig. 1 gives an overview of the data splits over training, validation, and testing.

Following previous imputation work, we work with signals in the form of $-\log_{10} p$-value tracks, which indicate the statistical significance of a mark at each genomic position, and seek to impute the average $-\log_{10} p$-value within each non-overlapping 25 base pair interval in a given subset of the genome. We additionally preprocess the $-\log_{10} p$-value signal using an *arcsinh* transform, which reduces the impact of outliers and differences in distribution between different types of assay, again inspired by prior work[22,23,42].

The EN-TEx dataset contains the results of a variety of functional genomic assays in 25 tissue types from four donors (in the main text, for consistency with prior publications, we use 'tissues' to refer to biosamples in EN-TEx and 'cell type' to refer to biosamples in Roadmap; for the purposes of the imputation method the two terms should be treated as interchangeable). We selected 116 histone modification tracks common to all four individuals (Supplementary Table 3). These tracks were processed in the same manner as those from Roadmap. The tracks measured for the 'thoracic aorta' and 'ascending aorta' were merged to cover all four individuals.

### Enrichment detection and evaluation metrics

We used MACS2[28] to detect peaks in observed tracks, using a one-sided Poisson hypothesis test with Benjamini-Hochberg correction for multiple test corrections and a cut-off value of 0.01. We refer to the 25-bp genomic bins belonging to the peaks detected by MACS2 for a given track as 'enriched bins' or 'foreground regions' for that track. Enriched bins detected in this way were used to define evaluation metrics for the tasks of both reference epigenome imputation and individual-specific imputation, as described in detail below and in Supplementary Section 3.2.

**Roadmap imputation metrics.** The global quality of imputations was measured using the mean squared error (MSE) and Pearson correlation coefficient applied to imputed and ground-truth tracks. These metrics were also evaluated separately on foreground and background regions. Recovery of enriched bins (i.e., bins occurring in MACS2 peaks) was measured using the threshold-agnostic area under the precision-recall curve (AUPRC). Finally, MACS2 was applied to imputed tracks to generate a set of predicted peaks using the same fixed parameters as used to call peaks on the observed tracks. The resulting peaksets were then compared with the peaksets returned from the observed tracks using precision and recall.

**Individual-specific imputation metrics.** Global imputation performance was measured as above. For the prediction of individual-specific peaks, we used the AUPRC to compare imputed tracks and MACS2 peaks, after excluding all genomic regions containing peaks conserved across individuals involved in the transfer learning and spurious individual-specific peaks of less than 150bp.

### Tensor factorization

Given a set of observed tracks that are the result of performing at least one of a set of $n_a$ assays $(a_1, \ldots, a_{n_a})$ in each of a set of $n_c$ cell types $(c_1, \ldots, c_{n_c})$, the goal is to generate imputations for all assay-cell type combinations which are not represented by tracks in the observed set. The complete set of possible measurements (all assays in all cell types at all genomic locations) can be represented as a rank-3 tensor $\mathcal{Y}$, with $\mathcal{Y}_{ijk}$ the signal observed at the $k^{th}$ genomic position when performing the $j^{th}$ assay in the $i^{th}$ cell type.

Tensor factorization approaches model entries in the tensor as interactions between separate representations for each dimension. In PREDICTD and Avocado, learned cell type embeddings, **c**, assay type embeddings, **a**, and genomic bin embeddings, **b**, are combined via a parametric function $g_\theta$ to reconstruct or impute tensor elements:

$$\hat{\mathcal{Y}}_{ijk} = g_\theta(\mathbf{c}_i, \mathbf{a}_j, \mathbf{b}_k). \tag{1}$$

The embeddings are learned to optimally reconstruct the observed tensor entries. Crucially, the use of a factorized functional form allows such models to generate predictions for arbitrary combinations of cell-type, assay and genomic location, meaning that

missing values in the tensor can be straightforwardly imputed given the learned embeddings.

## eDICE model

Given that individual epigenomic tracks are either completely observed or completely missing, to impute a particular missing entry $\mathcal{Y}_{ijk}$ corresponding to the signal value in a missing track at a particular genomic location $k$, the most important source of information is the observed values of other tracks at the same location. Let $Y^k$ represent the partially observed $(n_c \times n_a)$ matrix corresponding to taking a slice of the tensor at a particular genomic position. Our strategy is to learn to impute masked subsets of entries in $Y^k$ given the remaining entries, by learning a factorized regression function:

$$\hat{Y}_{ij}^k = g_\theta\left(\mathbf{c_i}(\tilde{Y}^k), \mathbf{a_j}(\tilde{Y}^k)\right). \tag{2}$$

Here $\tilde{Y}^k$ is the matrix of local signal values in which a subset of tracks has been masked by setting the corresponding entries of the matrix $Y^k$ to 0. All missing tracks likewise have their values set to 0. Factorization is achieved by encoding the matrix of local signal values into cell-type- and assay-type- specific representations, $\mathbf{c}_i(\tilde{Y}^k)$ and $\mathbf{a}_j(\tilde{Y}^k)$. These representations are thus directly conditioned on the local signal, unlike in the case of tensor factorization approaches, where cell-type and assay representations are global and parameterized directly.

The model produces the local embeddings $\mathbf{c}(\tilde{Y}^{(k)})$ and $\mathbf{a}(\tilde{Y}^{(k)})$ via separate cell and assay encoders. First, these encoders embed the local signal in each cell and each assay, then use self-attention to produce cell and assay embeddings that are informed by the signal in related cells and assays, respectively.

The model is trained by minimizing the mean squared error of the predictions of signal values in masked tracks in expectation over masks. The loss for a single genomic location is then:

$$\mathcal{L}(\theta, \phi, Y^k) = \frac{1}{|M|} \sum_{\{ij\} \in M} (\hat{Y}_{ij}^k(\theta, \phi) - Y_{ij}^k)^2. \tag{3}$$

The mean squared error is minimized with respect to the parameters of the encoders $\phi$ and decoder $\theta$ over a fixed training set of randomly selected genomic locations. At each iteration, a single mask $M$ is drawn at random for each location and used to compute a Monte Carlo estimate of the loss. In practice, we mask 120 tracks at a time ($|M| = 120$) and use the remaining tracks as 'context' to predict the masked values. This training objective can be seen as a kind of self-supervised learning, similar to that employed by denoising autoencoders[43], but differing in the use of a factorized encoder and decoder. At test time, all tracks from the training set are used as inputs to predict the values of held out tracks.

**Cell encoder.** Let $\mathbf{y}_{c_i}^{(k)}$ denote a partially observed signal vector characterizing the signal in tracks across all assays in cell type $c_i$ in the $k^{th}$ bin (i.e., the size of this vector is $n_a$, where $n_a$ is the total number of assays, some of which may be missing, and therefore set to 0 for the cell type in question). This cell-specific local signal vector is mapped to an embedding space through a non-linear function $\mathbf{f}_{\phi_C}$, shared by all cell types, and implemented through a fully connected layer with parameters $\phi_C$ and a ReLU activation function. To allow the network to combine the local signal representation with knowledge of the global properties of the cell type, we add to the local signal embedding a learned global cell type embedding $\mathbf{u}_c$, which plays the role of a position embedding in the standard Transformer architecture.

$$\mathbf{h}_{c_i}^k = \mathbf{f}_{\phi_C}(\mathbf{y}_{c_i}^k) + \mathbf{u}_{c_i} \tag{4}$$

The cell encoder then applies a Transformer-style self attention block to the resulting embeddings:

$$\mathbf{c}_1(\tilde{Y}^k), \ldots, \mathbf{c}_{n_c}(\tilde{Y}^k) = \text{SAB}(\mathbf{h}_{c_1}^k, \ldots, \mathbf{h}_{n_c}^k) \tag{5}$$

The self-attention block (SAB) is identical to a standard self-attentive Transformer layer[44], except for the removal of Layer Normalisation, which we did not find important in our shallow networks.

To account for differences in the number of observed entries across cell types, a scaling step is applied in the signal embedding. This step involves multiplying the activations of the fully connected layer $\phi_C$ by a factor $\frac{1}{n_{obs}}$, where $n_{obs}$ are the number of observed assays in the cell type, in an attempt to account for the uneven mapping of the epigenome, similar to the activation scaling used in Dropout[45].

**Assay encoder.** The assay encoder operates analogously to the cell encoder, taking as inputs assay signal vectors whose entries are the local signal values observed when performing a given assay in each cell type:

$$\mathbf{h}_{a_j}^k = \mathbf{f}_{\phi_A}(\mathbf{y}_{a_j}^k) + \mathbf{u}_{a_j} \tag{6}$$

$$\mathbf{a}_1(\tilde{Y}^k), \ldots, \mathbf{a}_{n_a}(\tilde{Y}^k) = \text{SAB}(\mathbf{h}_{a_1}^k, \ldots, \mathbf{h}_{a_{n_a}}^k). \tag{7}$$

**Signal Decoder.** The result of the factorized self-attention is a set of cell representations $(\mathbf{c_1}^k, \ldots, \mathbf{c}_{n_c}^k)$ and a set of assay representations $(\mathbf{a_1}^k, \ldots, \mathbf{a}_{n_a}^k)$, each of which is a function of the identity of the particular entity being represented and the full set of local signal values in all observed tracks at the $k$-th genomic bin ($\mathbf{c}_i^k \equiv \mathbf{c}_i(c_i, Y_{obs}^{(k)})$ and $\mathbf{a}_j^k \equiv \mathbf{a}_j(a_j, Y_{obs}^{(k)})$). Given these representations, the prediction for a given cell type-assay pair is obtained by passing the corresponding contextual cell type and assay representations through the fully connected neural network $g_\theta$ (Eq. (2)).

## Hyperparameters and training details

The model uses cell and assay embeddings of dimension 256 at all stages in processing. Within the self-attention block, we use 4 attention heads, whose output is concatenated and fed to a feed-forward neural network with a single hidden layer with 128 neurons and a 256-dimensional output. Finally, the combination of cell and assay representations is fed to a multilayer perceptron with 2 hidden layers with ReLU activations and 2048 neurons per layer. During training, Dropout with a rate of 0.3 is applied to each hidden layer in the output MLP.

The model used to analyse the eDICE performance in the Results section was trained on the union of the training and validation set for 50 epochs, using the Adam optimizer with a learning rate of $3 \times 10^{-4}$, and masking 120 randomly selected tracks to use as imputation targets for each training bin. Hyperparameters for this model were manually adjusted to maximise performance on the validation set.

For the EN-TEx imputations, the reconstruction task is modified so that the masked values belong to the same cell type in each individual bin, which closer mimics the generalization task analysed. The EN-TEx models have a reduced number of parameters in the embedding layers (128-dimensional) and the MLP hidden layers (512-dimensional), to account for the smaller dataset size. The transfer learning procedure involves training on one individual for 30 epochs, followed by 15 epochs of fine-tuning on the target individual with a reduced learning rate of $3 \times 10^{-5}$.

## Baselines

ChromImpute and PREDICTD imputations were downloaded directly from the resources accompanying their respective publications[21,22]. In the case of ChromImpute, these imputations were generated in a leave-one-out manner, while PREDICTD's imputations for tracks in our

test set were generated by models respecting the same train-test split used to train eDICE, and thus directly comparable to our results. Avocado's publicly available imputations, on the other hand, were generated by a model trained on the full Roadmap dataset (i.e., on all tracks, including the tracks in our test set), and therefore cannot be used to compare performance with other models. We, therefore, retrained an Avocado model from scratch to respect the data splits used here. To achieve this, we followed the two-stage procedure from[23], first training all parameters on chromosome 4, then freezing all parameters other than the genomic location embeddings, and fitting these for chromosome 21, to allow the generation of predictions for the test tracks on this chromosome. All results for Avocado refer to imputations made using this re-trained model.

### Reporting summary
Further information on research design is available in the Nature Portfolio Reporting Summary linked to this article.

## Data availability
The Roadmap dataset is available at http://www.roadmapepigenomics.org/. The epigenomic tracks for the 4 individuals part of the EN-TEx dataset can be found on the portal for the ENCODE project https://www.encodeproject.org/. The accession codes used for the EN-TEx analysis are listed in the Supplementary Material. The processed HDF5 files containing the training bins and chromosome 21 for the Roadmap dataset, and chromosome 21 for the selected tracks of the EN-TEx dataset can be found online at on Edmond, the open research data repository of the Max Planck Society[46]. Source data are provided with this paper.

## Code availability
Source code for eDICE[47] can be found at https://github.com/alex-hh/eDICE.

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

## Acknowledgements

This project has received funding from the Academy of Medical Sciences, UK (Springboard Fellowship SBF004/1060) and from UK Research and Innovation (Future Leader Fellowship MR/T022620/1) for author GS. It has also been supported by the European Union's Framework Programme for Research and Innovation Horizon 2020 (2014-2020) under the Marie Skłodowska-Curie Grant Agreement No. 813533-MSCA-ITN-2018 (for author GV) and by the German Federal Ministry of Education and Research (BMBF): Tübingen AI Center, FKZ: 01IS18039B, and by the Machine Learning Cluster of Excellence, EXC number 2064/1 - Project number 390727645 (for author BS). AHH was in part supported by the EPSRC Grant EP/S021566/1. The computing infrastructure was partly provided by the BMBF-funded de.NBI Cloud within the German Network for Bioinformatics Infrastructure (de.NBI) (031A532B, 031A533A, 031A533B, 031A534A, 031A535A, 031A537A, 031A537B, 031A537C, 031A537D, 031A538A). The authors thank the International Max Planck Research School for Intelligent Systems (IMPRS-IS) for supporting Tanmayee Narendra. GS would like to thank Dr Hartmut Schweikert for constructive criticism and advice.

## Author contributions

A.H.H. designed and implemented the model "imp" for the ENCODE imputation challenge as well as its subsequent improvements eDICE, contributed to the performance validation on the Roadmap dataset and to the writing of the manuscript. G.V. implemented and performed the analysis of the EN-TEx dataset, contributed to the Roadmap validation, the differential peak analysis, the implementation of the transfer learning strategy and the writing of the manuscript. TN implemented the differential peak analysis and contributed to the writing of the manuscript. M.R.-C. contributed to the supervision of the ENCODE Challenge contribution. B.S. has contributed to the conception of the work. GS supervised the development of the ENCODE imputation challenge method imp and eDICE, participated in the analysis of the Roadmap imputation, the differential peak analysis and the EN-TEx imputations, and contributed to the writing of the manuscript.

## Competing interests

The authors declare no competing interests.
