## [Peer Review File · Nature Communications]

Getting Personal with Epigenetics: Towards Individual-specific Epigenomic Imputation with Machine LearningREVIEWER COMMENTS

Reviewer #1 (Remarks to the Author):

This work proposes a deep learning framework using “local” cell type and embeddings for epigenetic data imputation, which, combined with self-attention, enables more compact and information-rich representations of cellular conditions and natures of the epigenetic features. The performance is in general better than previous models, which proves the design to be effective.

The authors experimented with several metrics for the evaluation of the imputation results. As experimental data are susceptible to systematic biases and random noises, it is difficult to directly compare different assays and cell types. These metrics could provide guidelines for assessing the potential usage of the imputation results. For example, capturing patterns and separating peaks and non-peaks would be more useful than accurately reconstructing the exact peak height; re-constructing patterns in active areas would be more important than the whole genome in general.

The authors also made comprehensive analysis for the differential performance between cell types and assays, which agrees with previously known biological knowledge. Another important observation is that personal information still needs to be included to precisely identify individual-specific signatures, even though cell type-specific patterns in general are mostly conserved across individuals.

This work offers some novel insights to not only the epigenetic data imputation task, but also to deep learning on epigenetic data in general. One major improvement of this study compared to previous factorization methods (like PREDICTD and Avocado) is the locality of cell type and assay embeddings, which combinatorically encode the epigenetic properties of a given genomic bin. As a result, there is no need for an additional “genomic position” embedding. This decreases the number of embeddings required and could potentially improve model interpretability.

The differential imputation performance between different epigenetic marks could indicate differences in their shapes and cell type-dependency. It could provide guidelines for processing of such data and for using them in other machine learning tasks.

Also, the transfer learning framework, especially the one pre-trained with other “observed” individuals from ENTE_x, proves to be quite successful. This could potentially be used to efficiently generalize the model to unseen cell types and/or individuals with incomplete epigenetic information.

The conclusions are well-supported and no major flaws with data analysis, interpretation and conclusions. Data visualization is clear and straightforward. The data pre-processing and the definition of the machine learning task follow a well-established problem setting, and the analysis of the results are comprehensive.

Major comments / suggested revisions:

The implementation details of the model and evaluation methods are sufficient, but there are a few points that may need some further explanation:

(Page 20) The objective of the model training seems to be to minimize the global MSE (like in Avocado). This should be explicitly stated.

-(Page 20) The input signal is normalized based on the number of available observations, which might also impact the scale of the output. Is the final prediction of the model adjusted accordingly?

-(Page 20) It seems the self-attention is only applied to the “cell type” and “assay” dimensions for each genomic position, while not along the genomic position dimension. However, the use of the term “contextualized” seems to refer to the latter scenario. Is there a reason for not aggregating information from neighboring genomic bins?

Minor comments:

Figure captions and citations need to be fixed in some places. For example:

- (Page 8) Fig 2f is not cited. It should be cited in the discussion of repressive and active marks. The broad and narrow peak comparison in the same paragraph should cite 2e.
- (Page 11) The lower panels of Fig 3a and 3b were not explained in the captions.
- (Page 13) Fig 4d is cited several times, but in some cases the text clearly refers to other subpanels.

Reviewer #2 (Remarks to the Author):

This manuscript presents a novel model, called eDICE, for epigenomic imputation. The model is more compact than state-of-the-art models and gives improved imputation performance in a previously established benchmark, using established performance measures as well as several novel performance measures. The manuscript then proceeds to apply eDICE to the ENTEX dataset, showing that the model can accurately predict histone modification profiles in individual donors.

The reduction in the number of parameters in the model is impressive. However, the critique of Avocado as requiring lots of memory is a bit strange. From the description, it sounds like Avocado requires memory equal to four times the length of the longest chromosome (because there are 100 genomic factors at 25bp resolution). This should easily fit in memory. It is not clear what is the benefit of reducing this footprint. Also, although the proposed model is indeed smaller, it requires that the training data be available at test time. So in a practical sense, using the model requires more space (either on disk or in memory) than Avocado. One might argue that reducing the total number of parameters is good because it avoids overfitting, but the paper does not provide any evidence that this is happening.

It seems unfortunate that Avocado was not included in the empirical comparison of methods, especially since the authors have introduced new performance measures. It should be possible to train an Avocado model on this dataset.

I was not particularly convinced by the section on capturing cell type specific differences. It's not really clear to me what this section adds relative to the previous section, with its array of performance measures. Furthermore, it's not clear why the differential analysis is only performed on eDICE and not competing methods.

Moving on to the sections about ENTEX, on the one hand, the ENTEX dataset is ripe for imputation, and having a completed dataset publicly available will be useful to the community. On the other hand, ENTEX is nearly complete, so the added value of imputation is minimal. In practical terms, imputation is useful to enable systematic analysis by machine learning or statistical analysis tools that do not handle missingness well (such as clustering or training a classifier). The manuscript focuses on the ability of their method to impute missing values (and specifically, differences between individuals) more accurately than baseline methods such as an averaging strategy. But in practice, the ENTEX data is mostly complete, so I would assume that most of these differences can be ascertained without requiring any imputation at all.

Overall, what is lacking here is a strong case for the utility of imputation in practice. The paper does not present any evidence that one can analyze an existing data set and, by making use of eDICE, learn things that were not apparent from directly analyzing the data itself.

The exploration of various training strategies for eDICE, including various transfer learning schemes, is only mildly informative. I don't really understand what the motivation for this exploration is, or what important lesson we learn from it.

Getting Personal with Epigenetics: Towards Machine-Learning-Assisted Precision Epigenomics (NCOMMS-22-04827)

Point-by-point response to the reviewers' comments

We thank both reviewers for their detailed feedback. We have sought to address all major comments and believe the resulting manuscript is significantly improved due to their constructive criticism. We provide point-by-point responses to individual comments below but highlight here a couple of notable changes. First, we have extensively revised the section on individual-specific imputation on the ENTE_x dataset, focussing on the task of predicting measurements in held-out tissues. Using this data set from four cadaveric donors, we can show that our model is able to transfer an understanding of data relatedness from tissue samples of the same individual onto data derived from a target individual. It provides a strong case for personalised imputation where certain tissues will require invasive biopsies while others are more readily probed. Second, we have added Avocado to our baselines in the section on Roadmap imputation, confirming that we outperform prior state-of-the-art imputation methods. Lastly, we provide learning curves which show that eDICE requires significantly less training data than its competitors.

Reviewer 1 (Remarks to the Author):

This work proposes a deep learning framework using “local” cell type and embeddings for epigenetic data imputation, which, combined with self-attention, enables more compact and information-rich representations of cellular conditions and natures of the epigenetic features. The performance is in general better than previous models, which proves the design to be effective. The authors experimented with several metrics for the evaluation of the imputation results. As experimental data are susceptible to systematic biases and random noises, it is difficult to directly compare different assays and cell types. These metrics could provide guidelines for assessing the potential usage of the imputation results. For example, capturing patterns and separating peaks and not the global MSE (like in Avocado-peaks would be more useful than accurately reconstructing the exact peak height; re-constructing patterns in active areas would be more important than the whole genome in general. The authors also made comprehensive analysis for the differential performance between cell types and assays, which agrees with previously known biological knowledge. Another important observation is that personal information still needs to be included to precisely identify individual-specific signatures, even though cell type-specific patterns in general are mostly conserved across individuals.

This work offers some novel insights to not only the epigenetic data imputation task, but also to deep learning on epigenetic data in general. One major improvement of this study compared to previous factorization methods

(like PREDICTD and Avocado) is the locality of cell type and assay embeddings, which combinatorially encode the epigenetic properties of a given genomic bin. As a result, there is no need for an additional “genomic position” embedding. This decreases the number of embeddings required and could potentially improve model interpretability. The differential imputation performance between different epigenetic marks could indicate differences in their shapes and cell type-dependency. It could provide guidelines for processing of such data and for using them in other machine learning tasks. Also, the transfer learning framework, especially the one pre-trained with other “observed” individuals from ENTE_x, proves to be quite successful. This could potentially be used to efficiently generalize the model to unseen cell types and/or individuals with incomplete epigenetic information.

RESPONSE: We thank the reviewer for a very positive summary of our main contributions and improvements relative to earlier methods.

The conclusions are well-supported and no major flaws with data analysis, interpretation and conclusions. Data visualization is clear and straightforward. The data pre-processing and the definition of the machine learning task follow a well-established problem setting, and the analysis of the results are comprehensive.

RESPONSE: Thank you!

Major comments / suggested revisions:

The implementation details of the model and evaluation methods are sufficient, but there are a few points that may need some further explanation:

1. (Page 20) The objective of the model training seems to be to minimize the global MSE (like in Avocado). This should be explicitly stated.

RESPONSE: Thank you for pointing this out. We have updated the manuscript accordingly and explicitly mention the objective in the methods section p. 14. of the updated draft.

2. (Page 20) The input signal is normalized based on the number of available observations, which might also impact the scale of the output. Is the final prediction of the model adjusted accordingly?

RESPONSE: We agree that normalization can affect the distribution of activations, but we found empirically that it has no harmful effect on the performance. We therefore do not adjust the final predictions.

3. Page 20) It seems the self-attention is only applied to the “cell type” and “assay” dimensions for each genomic position, while not along the genomic position dimension. However, the use of the term “contextualized” seems to refer to the latter scenario. Is there a reason for not aggregating information from neighboring genomic bins?

RESPONSE: Reviewer 1 is correct that in eDICE we are considering a given signal in the context of other measured signals in different assays and cell types. In early experiments, we found that incorporating information about genomic context did not improve prediction performance, although we agree that the idea is appealing, in particular with respect to a more interpretable model. However, the nature of ChIP-Seq measurements is such that each signal is measured genome-wide, such that an unmeasured target signal would also be missing in the neighbouring genomic bins. Therefore considering the genomic context would only provide additional information from off-target cell-type-assay combinations, which do not seem to add information relative to the same information on the target bin. We note that while ChromImpute and Avocado both incorporate information about genomic context, feature importance scores reported by the authors of ChromImpute indicate that it is the features at the target bin that are by far the most important. We hypothesise that the dependency of the missing values at the current bin on observed values of other tracks in surrounding bins is largely accounted for by the values of those tracks at the current bin. We have clarified our reasoning on page 13 of the revised manuscript.

Minor comments:

Figure captions and citations need to be fixed in some places.

RESPONSE: Thank you for pointing this out. We went over all figures, captions and their citations and removed all inconsistencies.

For example:

1. (Page 8) Fig 2f is not cited. It should be cited in the discussion of repressive and active marks. The broad and narrow peak comparison in the same paragraph should cite 2e.

RESPONSE: We made corresponding changes on page 5 of the revised manuscript. Thank you.

2. (Page 11) The lower panels of Fig 3a and 3b were not explained in the captions.

RESPONSE: Description of the aggregate plots added on page 24.

3. (Page 13) Fig 4d is cited several times, but in some cases the text clearly refers to other subpanels.

RESPONSE: We have fixed the references to sub panels of Figure 4.

Reviewer 2 (Remarks to the Author):

This manuscript presents a novel model, called eDICE, for epigenomic imputation. The model is more compact than state-of-the-art models and gives

improved imputation performance in a previously established benchmark, using established performance measures as well as several novel performance measures. The manuscript then proceeds to apply eDICE to the ENTE_x dataset, showing that the model can accurately predict histone modification profiles in individual donors. The reduction in the number of parameters in the model is impressive.

RESPONSE: We thank reviewer 2 for pointing out the significant improvements of our methods.

However, the critique of Avocado as requiring lots of memory is a bit strange. From the description, it sounds like Avocado requires memory equal to four times the length of the longest chromosome (because there are 100 genomic factors at 25bp resolution). This should easily fit in memory. It is not clear what is the benefit of reducing this footprint.

RESPONSE: The reviewer makes a fair point that it is not straight forward to quantify the benefit of reducing the memory footprint with regard to competing methods. However, we note that the Avocado paper indeed acknowledged the memory requirements of their model as presenting an issue: "Avocado does not fit a single model to the full genome because the genome latent factors could not fit in memory" (Schreiber et al, 2020).

Also, although the proposed model is indeed smaller, it requires that the training data be available at test time. So in a practical sense, using the model requires more space (either on disk or in memory) than Avocado. One might argue that reducing the total number of parameters is good because it avoids overfitting, but the paper does not provide any evidence that this is happening.

RESPONSE: We agree with the reviewers that any comparisons of model requirements involve tradeoffs. From a practical point of view, our model significantly reduces training cost by conditioning on data rather than learning embeddings because rather than having to train on all genomic positions for which embeddings are desired, we are able to train on a small subsample of the genome, and then predict at any other position as long as the input data for *that position* are available. This is something Avocado is incapable of. Critically, this is achieved while also significantly improving performance, as addressed in response to the next point. We have updated the manuscript accordingly to more accurately reflect where we believe the practical strengths of our model versus Avocado lie. In particular, we have added Figure 2D, which shows that eDICE is able to outperform Avocado even when trained on several orders of magnitude less training data. This suggests that Avocado severely overparameterized the task at hand. Our efficiency also opens up compelling possibilities to speed up the training even further by selecting suitable genomic regions for training rather than random sampling. Such a task requires a more robust evaluation and is left for future work.

It seems unfortunate that Avocado was not included in the empirical compari-

son of methods, especially since the authors have introduced new performance measures. It should be possible to train an Avocado model on this dataset.

RESPONSE: We agree with the reviewer’s suggestion and have updated the experiments and the manuscript accordingly (Compare Figure 2a, Supplementary Figures S6, S9, S23, and Supplementary Tables S1). We trained an Avocado model with a setup as close as possible to the original paper, and reported its performance. As described by the authors, the results are comparable to PREDICTD.

I was not particularly convinced by the section on capturing cell type specific differences. It’s not really clear to me what this section adds relative to the previous section, with its array of performance measures. Furthermore, it’s not clear why the differential analysis is only performed on eDICE and not competing methods.

RESPONSE: While we appreciate the reviewer’s feedback, we do believe that the section covering cell-type specific differences is important and adds value to the assessment of imputation methods. In the previous section, the prediction performance was estimated with regard to how well each predicted track individually and independently recapitulates the respective measured tracks. In most genomic bins, this task is facilitated by the fact that the signals are indeed correlated between many different cell types (see, for instance, Figure 4A Inlet.) However, there are also cell-type specific peaks, which are both harder to identify experimentally, but also to predict computationally. We argue that, ultimately, the concordance of these differences between measurements and predictions is the most relevant feature of a predictive tool. It is common practice to use statistical hypothesis testing in computational pipelines to analyse the experimental data, and therefore the same pipeline is used here for the predictions. We find good agreement between measurements and predictions in this particularly difficult task. We agree with the reviewer that the analysis should also be done with regard to the other predictive methods, and we have thus added this additional analysis Supplementary Fig. S23

Moving on to the sections about ENTE_x, on the one hand, the ENTE_x dataset is ripe for imputation, and having a completed dataset publicly available will be useful to the community. On the other hand, ENTE_x is nearly complete, so the added value of imputation is minimal. In practical terms, imputation is useful to enable systematic analysis by machine learning or statistical analysis tools that do not handle missingness well (such as clustering or training a classifier). The manuscript focuses on the ability of their method to impute missing values (and specifically, differences between individuals) more accurately than baseline methods such as an averaging strategy. But in practice, the ENTE_x data is mostly complete, so I would assume that most of these differences can be ascertained without requiring any imputation at all. Overall, what is lacking here is a strong case for the utility of imputation in practice.

RESPONSE: The ENTE_x data matrix shows that there are indeed many tis-

sues for which several assays remain missing; hence imputation is a valuable proposition for this data set. However, the main reason why this data set contains such a wide variety of tissues in the first place is that it is derived from cadaveric donors. In practice, many of these tissues are more difficult to sample in living individuals, and imputations are, therefore, important alternatives. In this study, we are most interested in exploring the possibilities and challenges of individualised imputation. We have revised this section and entirely dedicate it to imputing epigenomic signals for unobserved tissues by making use of transfer learning between individuals. The results of this task offer more direct applications for precision medicine. The availability of the ENTEEx dataset in its existing form is an extremely valuable resource for benchmarking the performance of methods on imputation of individual epigenomic data. We agree with the reviewer that explicitly making a strong case for the need for imputation methods further improves our manuscript. We have now added the following to the introduction: "Imputation models offer exciting new opportunities, and possible applications could lead to the development of novel precision medicine workflows; for instance, many tissues are difficult to probe in living patients, while a few others, such as blood samples, are readily accessible. We suggest that in the future computational models could be used to predict individual-specific epigenomic marks across tissues given measurements from accessible tissues alone, thus reducing the need for invasive biopsies."

The paper does not present any evidence that one can analyze an existing data set and, by making use of eDICE, learn things that were not apparent from directly analyzing the data itself.

RESPONSE: We agree that this paper's main contributions are not concerned with direct discovery of novel biology from imputed data. However, we feel that this is out of scope for the current work, which instead presents a methodological advance. Previous work, as well as the recent ENCODE imputation competition has demonstrated the utility of and need for imputation methods. It is reasonable to assume that improvements in these imputation methods are therefore of interest in themselves. Here, for the first time we present a method that is capable of individual-specific imputations. However we agree that the case for the interest of novel imputation methodology should be made more strongly and we have updated the manuscript accordingly.

The exploration of various training strategies for eDICE, including various transfer learning schemes, is only mildly informative. I don't really understand what the motivation for this exploration is, or what important lesson we learn from it.

RESPONSE: To the best of our knowledge strategies for individual-specific imputations have not been explored prior to this work. This setting differs from imputing "reference epigenomes" in two ways. Firstly, the number of available tracks per individual is much smaller, which makes it a harder problem. Secondly, in contrast to the reference epigenome where samples have been generated in different labs and from different genomic backgrounds, here tissue samples

from the same individual share an additional level of relatedness, beyond their tissue identity. Exploiting this relatedness in an efficient training strategy will be crucial for accurate individual-specific predictions in unseen tissues.

REVIEWERS' COMMENTS

Reviewer #1 (Remarks to the Author):

The authors have addressed all concerns from my previous review. I don't have any further comments and recommend publication of the manuscript.

Reviewer #2 (Remarks to the Author):

Thanks for addressing my various comments. I'll respond to them in the order they appear in the response to reviewers. Overall, I have only a few, relatively minor remaining concerns.

First, with respect to memory requirements, the quoted sentence from the Avocado paper seems to be explaining why the model is not set up to load the full genome into memory at once. It thus still seems funny for you to say, as your primary critique of Avocado, that "the learning of independent factors for each genomic location poses significant memory demands." In practice, it sounds like Avocado doesn't really need a lot of memory, but it would do so if it had been designed to work on the full genome at once. It seems to me that the problem you point out with Avocado is that there are too many parameters to be learned effectively. Indeed, in the second paragraph of your response, you say "From a practical point of view, our model significantly reduces training cost by conditioning on data rather than learning embeddings because rather than having to train on all genomic positions for which embeddings are desired, we are able to train on a small subsample of the genome, and then predict at any other position as long as the input data for that position are available." To me, something along these lines seems like what you should say in the manuscript, rather than incorrectly claiming that Avocado needs lots of memory. In particular, Figure 2D is a nice addition that directly addresses this point. You could add the claim that Avocado has too many parameters to the paragraph in question.

Thank you for including Avocado in the empirical comparison. It seems to me anyway like this makes the story more complete.

I was convinced by your arguments w.r.t. leaving in the section on tissue-specific imputations. Please be sure to discuss in the main text that you did this analysis for the other methods, and reference Supplementary Figure S23. Incidentally, I think panel C in that figure should be four small upset plots, rather than Venn diagrams.

I like the pivot from simply filling in the ENTE_x dataset to doing a case study on predicting individual epigenomes.

Typo on p. 7: inherit -> inherent

REVIEWERS' COMMENTS

Reviewer #1 (Remarks to the Author):

The authors have addressed all concerns from my previous review. I don't have any further comments and recommend publication of the manuscript.

RESPONSE : We thank the reviewer for the fruitful feedback provided, that allowed us to work to improve the overall clarity of our manuscript.

Reviewer #2 (Remarks to the Author):

Thanks for addressing my various comments. I'll respond to them in the order they appear in the response to reviewers. Overall, I have only a few, relatively minor remaining concerns.

First, with respect to memory requirements, the quoted sentence from the Avocado paper seems to be explaining why the model is not set up to load the full genome into memory at once. It thus still seems funny for you to say, as your primary critique of Avocado, that "the learning of independent factors for each genomic location poses significant memory demands." In practice, it sounds like Avocado doesn't really need a lot of memory, but it would do so if it had been designed to work on the full genome at once. It seems to me that the problem you point out with Avocado is that there are too many parameters to be learned effectively. Indeed, in the second paragraph of your response, you say "From a practical point of view, our model significantly reduces training cost by conditioning on data rather than learning embeddings because rather than having to train on all genomic positions for which embeddings are desired, we are able to train on a small subsample of the genome, and then predict at any other position as long as the input data for that position are available." To me, something along these lines seems like what you should say in the manuscript, rather than incorrectly claiming that Avocado needs lots of memory. In particular, Figure 2D is a nice addition that directly addresses this point. You could add the claim that Avocado has too many parameters to the paragraph in question.

RESPONSE: We thank the reviewer for clarifying this point. The mention of the memory cost of Avocado was removed. While the model PREDICTED did suffer from considerable constraints that required complex procedures, Avocado already offered significant improvements on that front. Instead, we emphasized that the tensor factorization models appear to significantly overparameterize the imputation problem, an issue that eDICE addresses by conditioning on data rather than learning explicit factors.

Thank you for including Avocado in the empirical comparison. It seems to me anyway like this makes the story more complete.

I was convinced by your arguments w.r.t. leaving in the section on tissue-specific imputations. Please be sure to discuss in the main text that you did this analysis for the other methods, and reference Supplementary Figure S23. Incidentally, I think panel C in that figure should be four small upset plots, rather than Venn diagrams.

RESPONSE : We mention that the differential analysis with DiffBind was repeated for all models considered, referencing supplementary Figure 23.

I like the pivot from simply filling in the ENTEX dataset to doing a case study on predicting individual epigenomes.

Typo on p. 7: inherit -> inherent

RESPONSE : Thank you. This has been corrected.

RESPONSE TO REVIEWERS' COMMENTS

We took the opportunity to work on improving the overall clarity of the presented work. The portions that were altered are highlighted in blue text; however, we emphasize the changes were not aimed at altering the content of the paper, but only to rephrase some portions to offer a clearer narrative to the reader. The only exception is the addition of three small sections in the Methods. The first, titled 'Enrichment detection and evaluation metrics', addresses the details of the peak calling procedure used. The sections titled 'Roadmap imputation metrics' and 'Individual-specific imputation metrics' offer a brief overview of the evaluation criteria used to measure the performance of the models.